# Probing the activated complex of the F + NH₃ reaction via a dipole-bound state

Rui Zhang [1], Shuaiting Yan[1], Hongwei Song [2] ✉, Hua Guo [3] & Chuangang Ning [1] ✉

Experimental characterization of the transition state poses a significant challenge due to its fleeting nature. Negative ion photodetachment offers a unique tool for probing transition states and their vicinity. However, this approach is usually limited to Franck-Condon regions. For example, high-lying Feshbach resonances with an excited HF stretching mode ($\nu_{HF} = 2$-$4$) were recently identified in the transition-state region of the $F + NH_3 \rightarrow HF + NH_2$ reaction through photo-detaching $FNH_3^-$ anions, but the direct photodetachment failed to observe the lower-lying $\nu_{HF} = 0,1$ resonances and bound states due apparently to negligible Franck-Condon factors. Indeed, these weak transitions can be resonantly enhanced via a dipole-bound state (DBS) formed between an electron and the polar $FNH_3$ species. In this study, we unveil a series of Feshbach resonances and bound states along the $F + NH_3$ reaction path via a DBS by combining high-resolution photoelectron spectroscopy with high-level quantum dynamical computations. This study presents an approach for probing the activated complex in a reaction by negative ion photo-detachment through a DBS.

Chemical reactions are always accompanied by the formation and cleaving of chemical bonds[1]. Right in the midst of this bond rearrangement, the system transforms from the reactant channel to the product channel through an activated complex that is called a transition state[2]. The transition state is typically located near a saddle point on the potential energy surface (PES) and is a key concept in chemistry for understanding chemical reactivity and kinetics[3,4]. As a result, an experimental characterization of the transition state has long been a holy grail in chemistry. However, it represents a significant challenge due to its transient nature. The motivation to better understand the transition state has spurred the development of the crossed molecular-beam technique[5,6], which has greatly improved our understanding of chemical dynamics[7-9]. In these experiments, the transition state is not directly probed, but its impact on reaction dynamics can be deduced from scattering attributes.

A complementary and more direct experimental method to characterize the transition state and its vicinity is the anion photoelectron spectroscopy[10-15]. Photodetachment of the anion projects its vibrational wavefunction vertically onto the neutral PES by ejecting the electron. If a stable anionic complex exists with a geometry similar to the transition state of the corresponding neutral reaction, the vibrationally resolved photoelectron spectrum contains detailed information on dynamics near the transition state[16,17]. Their assignment can be revealed by comparison with accurate quantum dynamical computations[18-21]. This technique is particularly suited for identifying metastable resonances, which are prevalent in many reactions and capable of impacting the reactivity[22]. It should be noted that a transition state is not a single geometry but an active complex in a region near the saddle point, often characterized by dynamical resonances[22].

Recently, Neumark and coworkers have demonstrated several exquisite experiments using the cryogenic slow electron velocity-map

[1]Department of Physics, State Key Laboratory of Low Dimensional Quantum Physics, Frontier Science Center for Quantum Information, Tsinghua University, 100084 Beijing, China. [2]State Key Laboratory of Magnetic Resonance Spectroscopy and Imaging, Innovation Academy for Precision Measurement Science and Technology, Chinese Academy of Sciences, Wuhan 430071, China. [3]Department of Chemistry and Chemical Biology, Center for Computational Chemistry, University of New Mexico, Albuquerque, NM 87131, USA. ✉e-mail: hwsong@wipm.ac.cn; ningcg@tsinghua.edu.cn

imaging method (Cryo-SEVI) to probe resonances spanning the transition-state region for F + H₂[23], F + CH₃OH[24], and F + NH₃[25]. Cryo-SEVI[15,26,27], featuring a very high energy resolution for slow photoelectrons, typically a few cm⁻¹ near the photodetachment threshold, is an excellent tool for resolving complicated vibrational structures. In these experiments, however, the observability of the vibrational structure on the neutral PES is limited to the Franck–Condon region, namely, only those with large overlaps with the anion vibrational wavefunction are detectable. In a recent cryo-SEVI experiment[25], for example, Babin et al. observed a series of peaks in the photoelectron spectrum of the FNH₃⁻ anion. With the help of quantum dynamics calculations, these peaks were assigned to Feshbach resonances with an excited HF stretching mode ($v_{HF}$ = 2–4) as well as several higher ones straddling the saddle point for the F + NH₃ → HF + NH₂ reaction. The absence of the lower-lying $v_{HF}$ = 0,1 resonances is presumably due to their negligible Franck–Condon factors.

A polar molecule with a dipole moment >2.5 D can support a dipole-bound state (DBS)[28,29], in which the extra electron is loosely bounded at the positively charged end through charge–dipole attraction, with a very low binding energy, typically ~a few tens of meV. The PES of the dipole-bound anion is nearly identical to the neutral PES since the neutral core is only slightly perturbed by the very diffusive DBS electronic wavefunction[30,31]. The loosely bound electron can serve as a messenger for the dynamic evolution of the neutral transition state[32]. Once the anion is excited to a DBS with a vibrationally excited neutral core, autodetachment occurs due to the vibronic coupling. This process can potentially enhance the observability of some vibrational states that may be too weak to be detected in direct

photodetachment. The existence of a DBS state also enables the pump-probe type experiment to observe ultrafast electronic dynamics, as recently demonstrated by Kim[33,34], Verlet[35–37], and their coworkers.

In this article, we present a joint cryo-SEVI and quantum dynamics study of the F + NH₃ → HF + NH₂ reaction, in which the low-lying resonances and bound states are detected via the DBS of the FNH₃⁻ anion. This work complements the earlier direct photodetachment experiment by Babin et al.[25] and provides a complete mapping of the transition-state region for the F + NH₃ reaction, which is a prototype for chemical reactions with a submerged reaction barrier[38].

## Results

The highly exoergic F + NH₃ → HF + NH₂ reaction has a submerged barrier flanked by the reactant complex (RC) and product complex (PC) wells. Figure 1 illustrates the schematic representation of how resonances in the transition-state region of the F + NH₃ reaction can be probed through the photoelectron detachment of FNH₃⁻ anions. The PC resonances with $v_{HF}$ = 2–4 have good Franck–Condon overlaps with the vibrational ground state of the anion, allowing for probing of these states via direct photodetachment. However, the overlaps with $v_{HF}$ = 1 states are quite weak, with roughly one order of magnitude smaller Franck–Condon factors (FCFs) than those of the $v_{HF}$ = 2 states. The overlaps with $v_{HF}$ = 0 states are even smaller. As a result, their signals in direct photodetachment are very weak, as noted in the recent work of Babin et al.[25]. However, DBS resonances can be photoexcited because they have a much larger cross-section than direct photodetachment, which can then autodetach, as depicted in Fig. 1. Additionally, the observation of these weak signals relies on the production of a strong FNH₃⁻ anion beam and the use of high-intensity photodetachment lasers.

To access the DBS, much longer photon wavelengths than those in the previous cryo-SEVI work ($hv$ > 27,000 cm⁻¹)[25] are used. At the lowest energy ($hv$ = 23,340 cm⁻¹), only a single peak (0a) is present. As the photon energy increases, more and more peaks (0b–0j) emerge. A key characteristic of DBS is the vibrational-state-specific enhancement depending on the photon wavelength. As shown in Fig. 2, the intensities of these peaks change dramatically as the photon energy increases, in contrast with the direct photodetachment, where the intensities of peaks are primarily governed by the FCFs and are proportional to $(E_k)^{1/2}$ for a s-wave Wigner threshold photodetachment[39]. Here, $E_k$ is the kinetic energy of photoelectrons. This characteristic clearly establishes the involvement of the DBS.

To confirm the existence of the DBS, we have performed coupled cluster calculations[40] for electron attachment. The dipole moment of FNH₃ was calculated using the density functional theory (DFT) with dispersion correction (see Supplementary Computational Methods for details). Our calculations showed the existence of a σ-type DBS with a binding energy of 327 and 415 cm⁻¹ for FNH₃ in its ground and $v_{HF}$ = 1 states, respectively, with the corresponding dipole moments of 4.8 and 5.0 D. The DBS electron locates mostly at the positively charged end –NH₂, as shown by the purple lobe of the corresponding molecular orbital in Fig. 1.

In Fig. 3, the experimental spectra are compared with the theoretical ones. The experimental spectrum in the low-energy wing, which is spliced from high-resolution parts of the multiple spectra in Fig. 2, represents the key finding of the current work. At higher energies, the current spectrum is in good agreement with that reported earlier by Babin et al.[25] dominated by direct photodetachment. The theoretical spectra are shifted 183 cm⁻¹ higher in binding energy to match the experiment. By comparing the experimental and theoretical spectra, the peaks can be assigned. In particular, the (0a–0j) and (1a–1f) bands correspond to bound and Feshbach resonance states associated with $v_{HF}$ = 0 and 1, respectively. However, the experimental intensity of the $v_{HF}$ = 0 band is roughly the same as that for $v_{HF}$ = 1, significantly different from the theoretical prediction based on direct

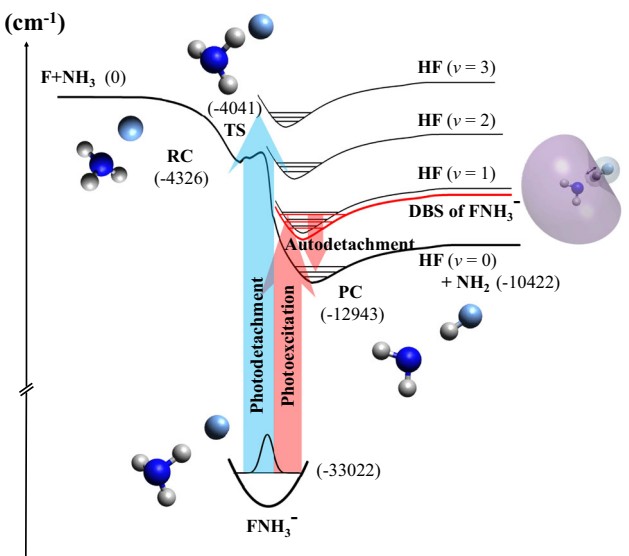

**Fig. 1 | Energy diagram for direct photodetachment and autodetachment via a dipole-bound state (DBS) of the FNH₃⁻ anion to the neutral F + NH₃ → HF + NH₂ reactive PES.** The upper bold curve represents the minimum energy path of the F + NH₃ reaction, connecting the reactants and products in their ground vibrational states. Three black curves sketched above it are neutral vibrational adiabatic potentials for the excited HF stretching mode ($v_{HF}$ = 1–3), and a red curve is for the dipole-bound state (DBS) of FNH₃⁻ anion with $v_{HF}$ = 1. The molecular orbital of the DBS is schematically plotted on a molecular structure next to the curve. Purple and light blue represent their different phases. The lower thick curve represents the anion PES. Structures around these curves are illustrated for the anion and the neutral reactant complex (RC), product complex (PC), and transition state (TS) in gray (H), dark blue (N), and light blue (F). The autodetachment of the DBS provides a bridge to access regions that are difficult to probe using the direct photodetachment, mainly due to the larger geometric difference between the anion and the neutral complex with $v_{HF}$ = 0. Energies shown in parentheses (in cm⁻¹) are zero-point energy corrected and are relative to the reactant asymptote.

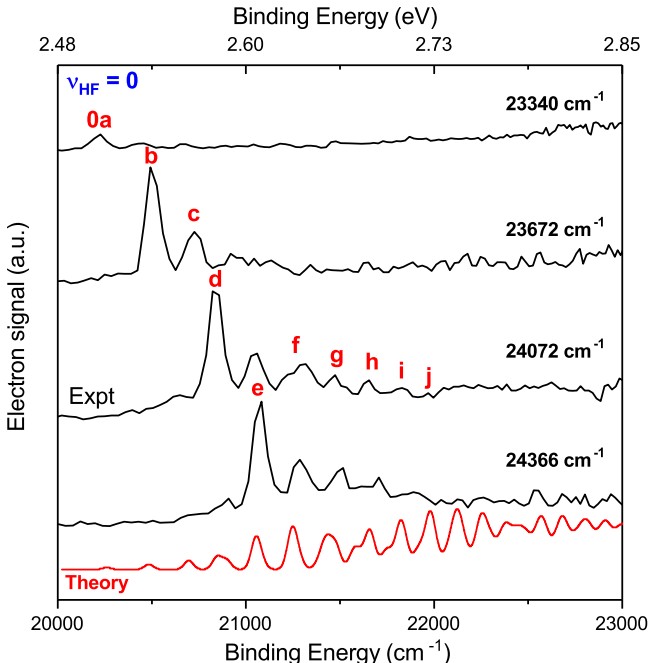

**Fig. 2 | Photoelectron energy spectra via the DBS autodetachment of FNH$_3^-$ with a vibrational-state-specific enhancement.** The intensity of peaks labeled 0a–0j depends sensitively on the photon energy due to the photoexcitation of different vibrational states of the DBS. The number 0 in the label represents all peaks observed in this figure assigned to the band $\nu_{HF} = 0$, which cannot be observed in the direct photodetachment experiment. Theoretical spectrum convoluted with a 50-cm$^{-1}$ FWHM Gaussian function (red) is shifted 183 cm$^{-1}$ higher in energy and plotted at the bottom for comparison.

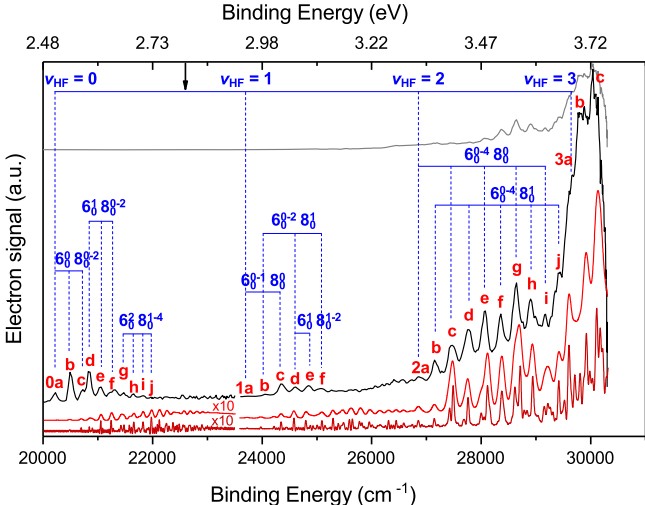

**Fig. 3 | Photoelectron energy spectra of FNH$_3^-$.** The top spectrum in gray represents the overview of experimental cryo-SEVI results. Below it, the high-resolution experimental spectra are shown in black. The raw theoretical spectrum (dark red) as well as the convoluted one (red, a 50-cm$^{-1}$ FWHM Gaussian function) are plotted for comparison at the bottom. The theoretical spectra are shifted 183 cm$^{-1}$ higher in the binding energy to match the experiment. The vertical dotted lines (blue) indicate the assignment of vibrational states. $\nu_6$ is the NH$_2$ out-of-plane wag mode, $\nu_8$ is the H$_2$N-HF stretching mode. The bold black arrow represents the product asymptote. The theoretical intensity of the band $\nu_{HF} = 0$ was multiplied by a factor of 10 for a better comparison. Numbers 0–2 in the labels represent the peaks assigned to the band $\nu_{HF} = 0$–2.

photodetachment. This is strong and additional evidence for the DBS enhancement of the $\nu_{HF} = 0$ detachment.

The band head peak labeled as 0a corresponds to the transition from the ground state of the FNH$_3^-$ anion to the ground state of the neutral FNH$_3$ complex. The experimental identification of the ground vibrational state of FNH$_3$ allowed an accurate determination of its electron affinity (EA) for the first time. Its experimental EA value was determined as 20,220(89) cm$^{-1}$, which agrees well with the theoretical prediction of 20,089 cm$^{-1}$. The fundamental frequency of the $\nu_{HF}$ mode in the FNH$_3$ complex was experimentally determined to be 3493 cm$^{-1}$ by comparing the positions of peaks 0a and 1a. This is also in excellent agreement with the theoretical result of 3470 cm$^{-1}$. Supplementary Table 1 compares experimental and theoretical peak shifts for FNH$_3^-$ detachment. The theoretical prediction of the shifts of the resonances agrees well with the experimental spectra. On the other hand, the predicted intensities of the $\nu_{HF} = 0$ resonances are in poor agreement with the experiment, even though the agreement for resonances in the $\nu_{HF} \geq 1$ bands is quite good, again underscoring the DBS-mediated mechanism in the former.

## Discussion

The large dipole moment of the FNH$_3$ complex (4.8 D) is sufficient to support a DBS. By tuning the photon energy, the FNH$_3^-$ anion in the ground state can be resonantly excited to the DBS. Furthermore, the binding energy of the DBS electron for $\nu_{HF} = 1$ (415 cm$^{-1}$) is significantly smaller than the vibrational energy. The vibrationally excited neutral core can autodetach due to vibronic coupling. This results in the ejection of the DBS electron, while the neutral core releases one vibrational quantum and returns to $\nu_{HF} = 0$, as depicted in Fig. 1. This photoexcitation process is often accompanied by the excitation of vibrational HF stretching mode ($\nu_{HF}$), NH$_2$ out-of-plane wag mode ($\nu_6$), and H$_2$N-HF stretching mode ($\nu_8$) due to the geometry changing.

Although the observed peak positions are well reproduced by theory, the anomalous intensities observed in the $\nu_{HF} = 0$ band, attributable to the resonant enhancement via the DBS, were not predicted by theory, as the vibronic autodetachment is not included in the theoretical treatment. An explicit simulation of autodetachment demands a quantum mechanical treatment of electron scattering beyond the scope of the current work.

Indeed, the spectra-profile dependence on the photon energy shown in Fig. 2 is likely due to the vibrational-state-specific photo-excitation and the mode competition during autodetachment. A variety of vibrational energy levels of the neutral core of the DBS can be reached when photoexciting the FNH$_3^-$ anion from its ground state by tuning the photon energy. When several modes were excited simultaneously, the mode that most significantly alters the dipole moment is more likely to be observed. Since the theoretical spectra in Fig. 3 simulate direct photodetachment, their intensities for peaks 0a–0j are understandably quite different from the experimental ones. To explore all possible photoexcitation resonance via the DBS, we conducted a scan of photon energy and recorded the electron signal and the anion beam signal. Interestingly, we did not observe any sharp resonances (refer to Supplementary Fig. 3). This absence of sharp resonances can presumably be attributed to a fast autodetachment process and dense vibrational peaks. In principle, some sharp DBS resonances may show up near the photodetachment threshold of $\nu_{HF} = 0$. However, we did not observe notable photoexcitation to the DBS with $\nu_{HF} = 0$, possibly due to the larger geometry difference from the FNH$_3^-$ anion. In the present work, the observed DBS resonances are well above the photodetachment threshold.

To gain a better understanding of the observed spectral features, vibrational wavefunctions of all the observed peaks were calculated. Since peaks 2a–2j have been investigated by Babin et al. in detail[25], we will focus here on the low-lying resonances (and bound states). In Fig. 4, we present the wavefunctions for peaks 0a–0j and 1a–1f, superimposed on the neutral PES. In panel 0a, we also include the contour of the anion wavefunction to illustrate their overlap during the

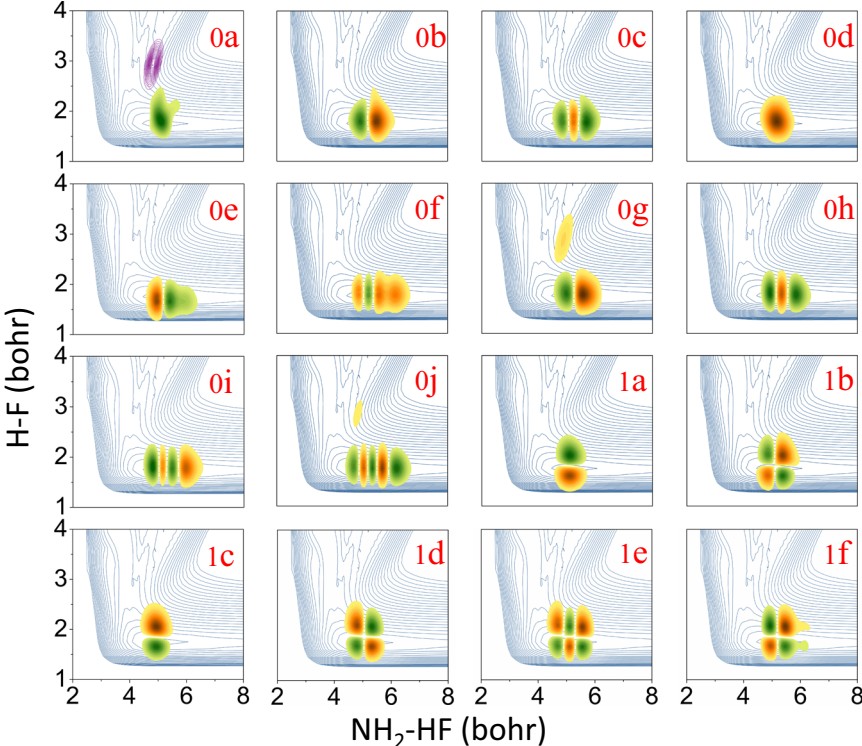

**Fig. 4 | 2D cuts of the resonance and bound state wavefunctions of the reaction F + NH$_3$ → HF + NH$_2$.** The labels correspond to the observed peaks in Figs. 2 and 3. Wavefunctions are superimposed on the neutral PES contour plotted with respect to the center-of-mass distance between NH$_2$ and HF ($R$) and H-F bond length ($r_1$), as defined in Supplementary Fig. 1. The purple contours in panel $0a$ represent the projection of anionic ground-state wave function onto the neutral PES. Peaks $0a$–$0j$ are related to the vibrational states with $v_{HF} = 0$, while peaks $1a$–$1f$ for $v_{HF} = 1$.

photodetachment. It can be seen that the overlaps between the anionic and neutral wavefunctions are generally very poor, resulting in difficulties in detecting them in the direct photodetachment experiment[25]. From the localized character of these wavefunctions, it is clear that they are metastable (presumably long-lived) resonances and bound states supported by the product well of the F + NH$_3$ reaction. Based on the product asymptote, states with energies below 22,600 cm$^{-1}$ are bound.

The assignment of these Feshbach resonances and bound states is confirmed by analyzing the nodal structure of the wavefunctions. Peaks $0a$–$0j$ exhibit no nodes along the HF stretching coordinate (vertical axis in Fig. 4), while peaks $1a$–$1f$ have one node, which is consistent with the assignment of $v_{HF} = 0$ and $v_{HF} = 1$. The node number along the horizontal axis in Fig. 4 indicates the vibrational quanta of the H$_2$N−HF stretching mode ($v_8$). For example, peaks $0a$ and $1a$ are assigned to $v_8 = 0$, while peaks $0c$ and $1e$ are $v_8 = 2$. The wavefunctions plotted along other coordinates are displayed in the Supplementary Fig. 2. All observed resonances can be assigned using the HF stretching mode ($v_{HF}$), NH$_2$ out-of-plane wagging mode ($v_6$), and H$_2$N−HF stretching mode ($v_8$). A detailed assignment is listed in Supplementary Table 1. The excitation of the $v_6$ mode is a result of the NH$_3$ pyramidal-to-planar transition, while $v_{HF}$ and $v_8$ modes arise from changes in the H−F and H$_2$N−HF equilibrium distances, respectively.

In summary, we have investigated the photodetachment of FNH$_3^-$ using cryo-SEVI spectroscopy and reduced-dimensional quantum dynamical computations on high-level ab initio PESs. We observed a new series of bound states and resonances for the prototypical F + NH$_3$ → HF + NH$_2$ reaction that was not previously reported, especially the series with the ground HF stretching ($v_{HF} = 0$) vibrational states, which are hard to detect in direct photodetachment. These neutral vibrational resonances/bound states with poor overlaps with the ground anionic state are detected through vibrational-state-

specific photoexcitation of a DBS of FNH$_3^-$ followed by autodetachment. Our high-level quantum dynamical calculations reproduced the energies of these bound and resonance states, allowing unambiguous assignment. Our results demonstrated a novel approach for probing the dynamics and spectroscopy of the transition-state region of a neutral reaction via a DBS, which is commonly present for polar species. It is worth noting that negative ions rarely have bound excited states; therefore, the existence of a DBS provides an opportunity for the time-resolved observations of reaction dynamics through pump-probe experiments. These transition-state spectroscopy studies are expected to provide a unique and complementary perspective of reaction dynamics to the molecular-beam approach.

## Methods
### Experimental Methods
The experiment was carried out on the Cryo-SEVI spectrometer[41,42]. In this work, a strong FNH$_3^-$ anion beam was generated by expanding a mixed gas of NF$_3$ and NH$_3$ (~1:2) through a pulsed valve fitted with a high-voltage discharge ion source. The backing pressure was around 6 × 10$^5$ Pa. The anions were guided by a radiofrequency (rf) hexapole guide and accumulated in a rf octupole ion trap held at 15 K[41] and cooled by collisions with the buffer gas (20% H$_2$ and 80% He) for 45 ms. After thermalization to their ground vibrational and electronic state, with only a few lower-lying rotational excited states, the anions were extracted from the trap and mass-selected by time-of-flight[43]. The anions were then photodetached at various photon energies with tunable light from an optical-parametric-oscillator (OPO) laser pumped by the third harmonic of a neodymium:yttrium−aluminum−garnet (Nd:YAG) laser. The outgoing photoelectrons were projected onto a microchannel-plate enhanced phosphor screen by the static electric field applied by a set of velocity-map-imaging (VMI) lens[44,45]. The electron-hitting positions were recorded via a charge-coupled device

(CCD) camera in an event-count mode and accumulated for typically 50,000 laser shots while running the spectrometer at a 20-Hz repetition rate. The photoelectron distribution was reconstructed from the projected imaging via the maximum entropy velocity Legendre reconstruction (MEVELER) method[46]. The VMI spectrometer features a high resolution for slow electrons, typically a few cm$^{-1}$ near the photodetachment threshold. A series of high-resolution spectra at different photon energies were concatenated to produce a full spectrum.

To search for the resonant photoexcitation, our spectrometer can be switched from the standard VMI mode to the scanning mode[47]. In the scanning mode, the phosphor screen was used as a charged particle detector. A high-speed oscilloscope was connected to the phosphor screen to record both the photoelectron signals and the residual anion signals after photodetachment on account of the different arriving times. To compensate for intensity fluctuations of the anion beam, the ratio of the intensity of the photoelectron signal to the intensity of the residual anion beam was monitored as a function of the scanned wavelength.

### Computational methods
The reactive system studied in this work is characterized by diatom–triatom (AB-CDE) Jacobi coordinates, as shown in Supplementary Fig. 1. Since the two NH bonds in the NH$_2$ moiety (N$_E$H$_C$H$_D$) keep mostly unchanged from the anion minimum to the neutral H$_2$N·HF well (from 1.019 to 1.021 Å at the UCCSD(T)-F12/aug-cc-pVTZ level)[48], they are expected to be spectators and should be nonactive in the process of photodetachment. Therefore, the lengths of the two non-reactive NH bonds are fixed at 1.93 $a_0$ in the dynamics calculations, i.e., $r_2$ and $r_3$ in the Jacobi coordinates are fixed at $r_{2e} = 1.97a_0$ and $r_{3e} = 1.93a_0$, respectively. Here $a_0$ is the Bohr radius. Within this constraint, the full-dimensional Hamiltonian is reduced to a seven-dimensional one that can be written as ($\hbar = 1$ hereafter)[49]:

$$
\hat{H} = -\frac{1}{2\mu_R}\frac{\partial^2}{\partial R^2} - \frac{1}{2\mu_{r_1}}\frac{\partial^2}{\partial r_1^2} + \frac{\left(\hat{J}_{tot} - \hat{j}\right)^2}{2\mu_R R^2} + \frac{\hat{j}_1^2}{2\mu_1 r_1^2} + \frac{\hat{l}_2^2}{2\mu_2 r_{2e}^2}
$$
$$
+ \frac{\hat{j}_3^2}{2\mu_3 r_{3e}^2} + \hat{V}(R, r_1, r_{2e}, r_{3e}, \theta_1, \theta_2, \theta_3, \varphi_1, \varphi_2), \qquad (1)
$$

where $R$, $r_1$, $r_2$, and $r_3$ are defined as the distance between centers of mass of H$_A$F$_B$ and H$_C$H$_D$N$_E$, the interatomic distance of H$_A$F$_B$, the distance from H$_C$ to the center of mass of H$_D$N$_E$, and the interatomic distance of H$_D$N$_E$, respectively, and the corresponding reduced masses are denoted by $\mu_R$, $\mu_1$, $\mu_2$, and $\mu_3$. $\hat{j}_1$ is the rotational angular momentum operator of H$_A$F$_B$, $\hat{l}_2$ is the orbital angular momentum operator along $r_2$, and $\hat{j}_3$ is the rotational angular momentum operator of H$_D$N$_E$. $\hat{J}$ is coupled by $\hat{j}_1$, $\hat{l}_2$ and $\hat{j}_3$.

The photodetachment of an anion can be reasonably approximated by the Condon model, which assumes that the electron detachment is much faster than nuclear motion. Within this approximation, the photoelectron spectrum is characterized by the squared overlap between the anionic wavefunction and the neutral scattering (or bound state) wavefunction. In this work, the initial wave packet is the ground vibrational state of the anion obtained by diagonalizing the seven-dimensional Hamiltonian on the anion PES. The wave packet is then propagated on the neutral PES using the Chebyshev propagator[50]. The energy spectrum is computed by cosine Fourier transforming the Chebyshev autocorrelation function. Further details of the quantum dynamics calculations can be found in Supplementary Computational Methods.

More than 41,000 energy points sampled in the relevant configuration space of the F + NH$_3$ reaction were calculated at the level of the unrestricted coupled cluster with singles, doubles, and perturbative triples with Dunning's augmented correlation-consistent polarized valence triple zeta basis set and core electrons frozen (UCCSD(T)-F12/

aug-cc-pVTZ)[48], and were then fitted by the fundamental-invariant neural network (FI-NN) method[51] to generate the globally accurate neutral PES. The anion PES was developed by fitting a total of ~11,000 energy points at the same level of theory as the neutral PES[25].

The dipole moment of FNH$_3$ was calculated using the dispersion-corrected double hybrid density functional theory at the level of B2PLYP-D3/aug-cc-pVTZ. The dipole moment of FNH$_3$ at its ground state was calculated using the equilibrium geometry, while the vibrationally averaged geometry was used to calculate the dipole moment for $v_{HF} = 1$. The DBS of FNH$_3$ was calculated using the equation of motion for electron attachment coupled cluster method with single and double excitations (EOM-EA-CCSD)[40]. To accurately describe the diffusive nature of DBS, extra diffusive functions 6s6p in an even-tempered manner were added to the standard aug-cc-pVTZ basis sets for each atom. Further details of DBS calculations can be found in Supplementary Computational Methods.

### Data availability
The raw spectral and calculated data of figures generated in this study have been provided in the figshare database and can be obtained at https://doi.org/10.6084/m9.figshare.25433896[52]. The measured data of all peaks and the numerical parameters used in the calculation in this study are provided in the Supplementary Information.

### Code availability
The associated codes, such as the subroutine to generate anion and neutral PESs and the quantum scattering code, are available on GitHub at https://github.com/apmtcc/AB-CDE[53] and described in the README file.

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

## Acknowledgements
We thank the National Natural Science Foundation of China (NSFC) (Grant Nos. 12374244 and 11974199 to C.N., 21973109 and 21921004 to H.S.). S.Y. thanks the China Postdoctoral Science Foundation (Grant no. GZC20231367). H.G. is supported by the US Air Force Office of Scientific Research (Grant no. FA9550-22-1-0350).

## Author contributions
The experiments were conceived by C.N. and carried out by R.Z. and S.Y. The data analysis was performed by R.Z. with support from C.N. The calculations were conceived by H.S. and H.G. and performed and analyzed by H.S. with assistance from H.G. The paper was written by R.Z., H.S., H.G. and C.N.

## Competing interests
The authors declare no competing interests.
