## [Peer Review File · Nature Communications]

Probing the activated complex of the $F + NH_3$ reaction via a dipole-bound stateREVIEWER COMMENTS

Reviewer #1 (Remarks to the Author):

This work provides the novel observation of quantum states corresponding to the ground or first-excited HF from the cryo-SEVI spectroscopy of the FNH₃(-) anion complex. Because of the rather large structural change of the neutral from the anionic complex, it had been considered to be not plausible to detect the low or ground HF quantum states from SEVI. In this work, authors tuned the photodetachment laser pulse so that it could access the metastable DBS species of which the autodetachment gives the photoelectron signals associated with the ground or (V=1) HF quantum states. Although Feshbach resonances of DBS were not obviously found in the photodetachment spectrum, the comparison with the theoretical FC simulation based on the wavefunction overlaps strongly supports the authors' interpretation. The mere fact that they could observe the more detailed quantum structures leading to the ground (or first-vibrationally excited) HF states is truly outstanding. The role of the DBS is quite reasonable though its dynamic property seems to be subject to the further investigation. Authors may want to tone down a little bit about the critical role of the DBS in the SEVI (Namely, is it also possible that authors got the more sensitive SEVI signal?). Otherwise, it merits the publication in Nature Comm.

Reviewer #2 (Remarks to the Author):

Zhang et al. identify so-far unreported bound and Feshbach resonance states corresponding to the HF($v = 0$ and 1) vibrational states near the post-reaction-complex region of the $F + NH_3 \rightarrow HF + NH_2$ reaction through the photoexcitation and subsequent autodetachment of the FNH₃⁻ anion. These states could not be determined experimentally via direct photodetachment due to their small Franck-Condon factors with the anion complex. Photoexcitation to the dipole bound states (DBSs) of the FNH₃⁻ complex, where the electron is only loosely attached to the neutral core, followed by autodetachment, allows for accessing the low-lying HF($v = 0$ and 1) resonance and bound states by providing larger cross sections than direct photodetachment. Based on accurate reduced-dimensional quantum dynamics calculations (where the two spectator NH bond lengths are fixed), simulating direct photodetachment, the unambiguous assignment of the experimentally found states can be done.

This work demonstrates the usefulness of a photoexcitation tool to probe regions of reactive potential energy surfaces that are out of the Franck-Condon-active areas, but might bear interesting features mediating the dynamics of a chemical reaction, by identifying so-far experimentally inaccessible resonance and bound states. The manuscript is well written and may capture the interest of a broad scientific community. Thus, I think this study deserves publication in Nature Communications after considering some minor comments:

The studied resonance and bound states do not seem to be in the transition-state region, but rather in the post-reaction-complex region of the PES. I suggest modifying the title (and slightly the abstract and the introduction) accordingly or the authors should elaborate on this issue. For example, a more transparent structure of the DBS could be shown in Figure 1 to be able to see its position along the reaction coordinate.

r20 and r30 in the Hamiltonian are very strange notations for the fixed coordinates.

j23 operator is defined in line 319, but the Hamiltonian does not contain this operator.

Line 439: CM should be spelled out.

Some typos:

In line 22: their vicinity

In line 195: "unambiguously" should be "unambiguous"

In SI line 184: "exponents ak was" should be "exponents ak were"

Reviewer #3 (Remarks to the Author):

Reviewer #4 (Remarks to the Author):

Negative ion photodetachment is a well-understood spectroscopic probe that is sensitive to the transition state regions of bimolecular reactions. One of its most impactful outcomes has been to enhance our understanding of reactive scattering through the characterization of quasi-bound collisional complexes (resonances) in the vicinity of such transition states. However, this approach can only access states that have a significant degree of Franck-Condon overlap with the anion being probed. The noteworthy aspect of this manuscript from Song, Ning, and coworkers is that they have demonstrated a complementary approach to negative ion photodetachment that enables additional states to be probed, provided that they can be accessed through a dipole-bound state (DBS) created from the initial anion. In principle this method has the potential to enhance our understanding of transition state chemistry by providing a new approach to potential energy surface mapping.

In fact, the authors demonstrate the complementarity of their technique by presenting their DBS-based measurements on the HF($v = 0,1$) states of the F + NH₃ reaction as building on prior work (involving two of the same authors) that used negative ion photoelectron spectroscopy to explore the HF($v = 2-4$) states. Indeed, it is hard to read this paper without having the other in mind (although this is not meant in any negative sense).

In general, I was impressed and convinced by the quality of the results presented. The authors provide several strong arguments for their claim that their measurements are sensitive to electron detachment following the creating of a DBS. These include the relative peak intensities, which are significantly different from those predicted for direct photodetachment, as well as the observed change in vibrational excitation as a function of photon energy. The authors also use their methodology to report on some physically interesting but previously unmeasured properties such as the transition state electron affinity and fundamental frequency. Their methodology is sound and enough information is provided for others to attempt to reproduce their results.

My only concern with the manuscript is that the authors do not really justify how this work will impact the field of reaction dynamics more broadly. Despite proving principle for DBS-based transition state measurements, it isn't clear how the restrictions involved (very large dipole moments, dipole-bound states with geometries similar to the neutral transition state) limit the applicability. Even so, this is a very strong paper that I would be happy to see in Nature Communications. I recommend it be published subject to minor revisions.

In addition to the above review, I also have some small questions/comments for the authors:

- Clarke and Verlet recently published a review on how gas-phase anions can be used to explore excited-state chemistry, which could be useful to cite here ("Dynamics of Anions: From Bound to Unbound States and Everything In Between." Annual Review of Physical Chemistry 75 (2024)).
- Figure 1: It would be helpful to superimpose the molecular orbital on the relevant structure (possibly it is, but the resolution makes it difficult to tell).
- Figure 2: Is the theoretical spectrum shifted in energy here, as it is in Figure 3?
- Lines 142-144: This reads a bit unclearly. I think what is meant is that the relaxation of the DBS (through the HF mode) can result in the product HF being vibrationally excited, but it reads as if the relaxation of one mode can be accompanied by its excitation.
- Line 145: The v_7 mode is not mentioned again in the results (either in the main text or the supporting material).

Re: NCOMMS-24-07358-T

Title: Probing the activated complex of the F + NH₃ reaction via a dipole-bound state

We appreciate the comments and suggestions from all the reviewers. In the revised manuscript, we have modified the title and some expression surrounding the “transition state region” and also added the content of the “dipole-bound state” in response to reviewers. The revisions are listed points-to-points below, and the revised files also include a version where all changes are marked in red.

Response to the comments.

Reviewer #1:

1) This work provides the novel observation of quantum states corresponding to the ground or first-excited HF from the cryo-SEVI spectroscopy of the FNH₃(-) anion complex. Because of the rather large structural change of the neutral from the anionic complex, it had been considered to be not plausible to detect the low or ground HF quantum states from SEVI. In this work, authors tuned the photodetachment laser pulse so that it could access the metastable DBS species of which the autodetachment gives the photoelectron signals associated with the ground or (V=1) HF quantum states. Although Feshbach resonances of DBS were not obviously found in the photodetachment spectrum, the comparison with the theoretical FC simulation based on the wavefunction overlaps strongly supports the authors' interpretation. The mere fact that they could observe the more detailed quantum structures leading to the ground (or first-vibrationally excited) HF states is truly outstanding.

Response: We thank the Reviewer's appreciation of the work.

2) The role of the DBS is quite reasonable though its dynamic property seems to be subject to the further investigation. Authors may want to tone down a little bit about the critical role of the DBS in the SEVI (Namely, is it also possible that authors got the more sensitive SEVI signal?). Otherwise, it merits the publication in Nature Comm.

Response: We appreciate the reviewer's comments. We agree that the successful observation of the weak bands is not solely attributed to the DBS. In the revised version (Line 96-97), we have included the following statement: “Additionally, the observation of these weak signals relies on the production of a strong FNH₃⁻ anion beam and the use of high-intensity photodetachment lasers.”

To generate a strong FNH₃⁻ anion beam, we employed a pulsed valve equipped with a high voltage discharge ion source to expand the mixed gas (NF₃/NH₃). (Line 283). This approach allowed us to observe resonances/bound states with $v_{\text{HF}} = 1$ through the direct detachment, even though its Franck-Condon factors (FCFs) roughly one order of magnitude smaller than those of the $v_{\text{HF}} = 2$ states.

It is worth noting that DBS indeed plays a crucial role in facilitating the observation the lower-lying $\nu_{\text{HF}} = 0$ resonances/bound states. Supporting evidence includes the significant changes in intensities of different features ($0a-0j$) as the photon energy increases, as well as the experimental intensity of the $\nu_{\text{HF}} = 0$ band being comparable to that of $\nu_{\text{HF}} = 1$. The observed intensities deviate significantly from the theoretical prediction based on direct photodetachment.

Reviewer #2:

1) Zhang et al. identify so-far unreported bound and Feshbach resonance states corresponding to the HF($v = 0$ and 1) vibrational states near the post-reaction-complex region of the $\text{F} + \text{NH}_3 \rightarrow \text{HF} + \text{NH}_2$ reaction through the photoexcitation and subsequent autodetachment of the FNH_3^- anion. These states could not be determined experimentally via direct photodetachment due to their small Franck-Condon factors with the anion complex. Photoexcitation to the dipole bound states (DBSs) of the FNH_3^- complex, where the electron is only loosely attached to the neutral core, followed by autodetachment, allows for accessing the low-lying HF($v = 0$ and 1) resonance and bound states by providing larger cross sections than direct photodetachment. Based on accurate reduced-dimensional quantum dynamics calculations (where the two spectator NH bond lengths are fixed), simulating direct photodetachment, the unambiguous assignment of the experimentally found states can be done.

This work demonstrates the usefulness of a photoexcitation tool to probe regions of reactive potential energy surfaces that are out of the Franck-Condon-active areas, but might bear interesting features mediating the dynamics of a chemical reaction, by identifying so-far experimentally inaccessible resonance and bound states. The manuscript is well written and may capture the interest of a broad scientific community. Thus, I think this study deserves publication in Nature Communications after considering some minor comments:

Response: We thank the Reviewer for the valuable feedback on our work and finding our work worthy of publishing in Nature Communications.

2) The studied resonance and bound states do not seem to be in the transition-state region, but rather in the post-reaction-complex region of the PES. I suggest modifying the title (and slightly the abstract and the introduction) accordingly or the authors should elaborate on this issue. For example, a more transparent structure of the DBS could be shown in Figure 1 to be able to see its position along the reaction coordinate.

Response: Thanks for the comments and suggestions. Strictly speaking, the resonant and bound states we observed in this work belong to the product-complex region, as shown in Figure 4. The “transition state region” in the original title is a generalized concept defined as the state between the reactant and the product, including the complex

at the product end. The manuscript also has a description in the Line 37-38 and Line 53-55. ('Right in the midst of this bond rearrangement, the system transforms from the reactant channel to the product channel through an activated complex that is called a transition state'; 'It should be noted that a transition state is not a single geometry, but an active complex in a region near the saddle point, often characterized by dynamical resonances.')

To avoid any possible misunderstanding, we accept Reviewer's suggestion and modify the title to be "Probing the **activated complex** of the F + NH₃ reaction via a dipole-bound state". The term of activated complex in the reaction includes the transition state and its vicinity. We have also modified several relevant languages in the abstract and introduction as you suggested.

In addition, we have updated a more transparent structure of the DBS with a scaled up molecular structure in Fig.1.

3) r_{20} and r_{30} in the Hamiltonian are very strange notations for the fixed coordinates.

Response: Yes, r_{20} and r_{30} are the fixed coordinates of the equilibrium structure. For clarity, we have changed them as r_{2e} and r_{3e} .

4) j_{23} operator is defined in line 319, but the Hamiltonian does not contain this operator.

Response: The operator \hat{j}_{23} was initially used to define the operator \hat{j} . We have changed the statement in Line 323 as " \hat{j} is coupled by \hat{j}_1 , \hat{l}_2 and \hat{j}_3 ".

5) Line 439: CM should be spelled out.

Response: "CM" is the abbreviation of "center of mass". We have spelled it out.

6) Some typos:

In line 22: their vicinity

In line 195: "unambiguously" should be "unambiguous"

In SI line 184: "exponents ak was" should be "exponents ak were"

Response: Thanks, we have fixed these typos.

Reviewer #3:

Response: Thanks.

Reviewer #4:

1) Negative ion photodetachment is a well-understood spectroscopic probe that is sensitive to the transition state regions of bimolecular reactions. One of its most impactful outcomes has been to enhance our understanding of reactive scattering through the characterization of quasi-bound collisional complexes (resonances) in the vicinity of such transition states. However, this approach can only access states that have a significant degree of Franck-Condon overlap with the anion being probed. The noteworthy aspect of this manuscript from Song, Ning, and coworkers is that they have demonstrated a complementary approach to negative ion photodetachment that enables additional states to be probed, provided that they can be accessed through a dipole-bound state (DBS) created from the initial anion. In principle this method has the potential to enhance our understanding of transition state chemistry by providing a new approach to potential energy surface mapping.

In fact, the authors demonstrate the complementarity of their technique by presenting their DBS-based measurements on the HF($v = 0,1$) states of the F + NH₃ reaction as building on prior work (involving two of the same authors) that used negative ion photoelectron spectroscopy to explore the HF($v = 2-4$) states. Indeed, it is hard to read this paper without having the other in mind (although this is not meant in any negative sense).

Response: Thanks for the comments. Indeed, the previous work from Babin et al. has provided the initial inspiration. This current measurements on the HF (\$v = 0,1\$ ) states based on DBS plus the earlier direct photodetachment experiment of HF (\$v = 2-4\$ ) states would provide a more complete mapping of the transition-state region for the F + NH₃ reaction.

2) In general, I was impressed and convinced by the quality of the results presented. The authors provide several strong arguments for their claim that their measurements are sensitive to electron detachment following the creating of a DBS. These include the relative peak intensities, which are significantly different from those predicted for direct photodetachment, as well as the observed change in vibrational excitation as a function of photon energy. The authors also use their methodology to report on some physically interesting but previously unmeasured properties such as the transition state electron affinity and fundamental frequency. Their methodology is sound and enough information is provided for others to attempt to reproduce their results.

Response: We greatly appreciate the valuable feedback provided by the Reviewer.

3) My only concern with the manuscript is that the authors do not really justify how this work will impact the field of reaction dynamics more broadly. Despite proving principle for DBS-based transition state measurements, it isn't clear how the restrictions involved (very large dipole moments, dipole-bound states with geometries similar to the neutral transition state) limit the applicability. Even so, this is a very strong paper that I would be happy to see in Nature Communications. I recommend it be published subject to minor revisions.

Response: Yes, there are limitations to the applicability of the DBS-based transition state measurement, such as requiring a dipole moment >2.5 D and a geometrical structure similar to the neutral transition state. Nevertheless, it provides a new approach to probe reaction dynamics.

In the revised version, we have included the following statement for the potential impact (Line 197-199):

“It is worth noting that negative ions rarely have bound excited states; therefore, the existence of a DBS provides an opportunity for the time-resolved observations of reaction dynamics through pump-probe experiments.”

In addition to the above review, I also have some small questions/comments for the authors:

4)- Clarke and Verlet recently published a review on how gas-phase anions can be used to explore excited-state chemistry, which could be useful to cite here ("Dynamics of Anions: From Bound to Unbound States and Everything In Between." Annual Review of Physical Chemistry 75 (2024)).

Response: Thanks for the suggestion. We have added this nice literature as Ref. 37 in the revised manuscript.

5)- Figure 1: It would be helpful to superimpose the molecular orbital on the relevant structure (possibly it is, but the resolution makes it difficult to tell).

Response: Thanks for the nice suggestion. The molecular orbital of the dipole bound state is very diffusive and its size is much larger than the molecular size. The molecular structure would not be visible if the real scale is used. We have adjusted the scale of molecular structure and dipole-bound molecular orbitals in Figure 1 for a better view. We also have noted that the ratio of that two is different from the real ratio in the caption which is just used for schematic. Thanks for the suggestion.

6)- Figure 2: Is the theoretical spectrum shifted in energy here, as it is in Figure 3?

Response: Yes. The theoretical spectra are shifted in the same energy 183 cm^{-1} in both Figures 2 and 3. In the revised version, we have specified the shift in the caption of Fig.2 in Line 425-426.

7)- Lines 142-144: This reads a bit unclearly. I think what is meant is that the relaxation of the DBS (through the HF mode) can result in the product HF being vibrationally excited, but it reads as if the relaxation of one mode can be accompanied by its excitation.

Response: We thank the Reviewer for pointing this out. The vibrational modes were excited in the DBS photoexcitation process. Then, the vibrational modes relaxed during the autodetachment. In the revised version (Lin142), we have replaced “photoexcitation/autodetachment” with “photoexcitation”.

8)- Line 145: The ν_7 mode is not mentioned again in the results (either in the main text or the supporting material).

Response: Yes. Both ν_6 or ν_7 are the NH_2 out-of-plane wag modes. But we only observed the features caused by ν_6 mode in the spectra based on the reduced-dimensional quantum dynamics calculations. Indeed, no features from the ν_7 mode have been observed in this work. To avoid misunderstanding, we have deleted it in the revised manuscript (Line 144).

Yours sincerely,

Chuangang Ning